

# Spatial heterogeneity of physicochemical properties explains differences in microbial composition in arid soils from Cuatro Cienegas, Mexico

Silvia Pajares[1,2], Ana E. Escalante[3], Ana M. Noguez[4], Felipe García-Oliva[5], Celeste Martínez-Piedragil[5], Silke S. Cram[6], Luis Enrique Eguiarte[4] and Valeria Souza[4]

[1] Instituto de Ciencias del Mar y Limnología, Unidad Académica de Ecología y Biodiversidad Acuática, Universidad Nacional Autónoma de México, Mexico City, Mexico
[2] Departamento de Procesos y Tecnología, Universidad Autónoma Metropolitana, Unidad Cuajimalpa, Mexico City, Mexico
[3] Instituto de Ecología, Laboratorio Nacional de Ciencias de la Sostenibilidad, Universidad Nacional Autónoma de México, Mexico City, Mexico
[4] Instituto de Ecología, Departamento de Ecología Evolutiva, Universidad Nacional Autónoma de México, Mexico City, Mexico
[5] Instituto de Investigaciones en Ecosistemas y Sustentabilidad, Universidad Nacional Autónoma de México, Morelia, Michoacán, Mexico
[6] Instituto de Geografía, Universidad Nacional Autónoma de México, Mexico City, Mexico

Corresponding author
Ana E. Escalante,
anaelena.escalante@gmail.com

## ABSTRACT

Arid ecosystems are characterized by high spatial heterogeneity, and the variation among vegetation patches is a clear example. Soil biotic and abiotic factors associated with these patches have also been well documented as highly heterogeneous in space. Given the low vegetation cover and little precipitation in arid ecosystems, soil microorganisms are the main drivers of nutrient cycling. Nonetheless, little is known about the spatial distribution of microorganisms and the relationship that their diversity holds with nutrients and other physicochemical gradients in arid soils. In this study, we evaluated the spatial variability of soil microbial diversity and chemical parameters (nutrients and ion content) at local scale (meters) occurring in a gypsum-based desert soil, to gain knowledge on what soil abiotic factors control the distribution of microbes in arid ecosystems. We analyzed 32 soil samples within a 64 m$^2$ plot and: (a) characterized microbial diversity using T-RFLPs of the bacterial 16S rRNA gene, (b) determined soil chemical parameters, and (c) identified relationships between microbial diversity and chemical properties. Overall, we found a strong correlation between microbial composition heterogeneity and spatial variation of cations (Ca$^2$, K$^+$) and anions (HCO$_3^-$, Cl$^-$, SO$_4^{2-}$) content in this small plot. Our results could be attributable to spatial differences of soil saline content, favoring the patchy emergence of salt and soil microbial communities.

## INTRODUCTION

Spatial heterogeneity is an inherent feature of soils and has significant functional implications, including the fact that different soil patches and aggregates can present variation in nutrient transformation rates (e.g., respiration, mineralization, nitrogen fixation) (*Noguez et al., 2008*; *Strickland et al., 2009*; *Zeglin et al., 2009*), particularly when the activities and distribution of microorganisms are considered. The scale at which environmental variation is considered in association with microbial diversity varies greatly, from tens to thousands of kilometers, to meters and even at the microscale (*Vos et al., 2013*). Depending on the spatial scale at which microbial diversity is studied, different environmental parameters and ecological processes may be associated to the observed diversity distribution (*Martiny et al., 2011*). At large spatial scales (tens to thousands of kilometers), soil microbial community structure is correlated to edaphic variables, such as soil pH (*Fierer & Jackson, 2006*), temperature (*Garcia-Pichel et al., 2013*), and moisture content (*Angel et al., 2010*). At smaller scales (tens of meters), plant communities have been shown to have a strong influence on soil microbial diversity through interactions within the rhizosphere (*Berg & Smalla, 2009*; *Hartmann et al., 2009*; *Ben-David et al., 2011*). However, little is known about the effects of small-scale habitat variation on the spatial patterns of microbial diversity and its interactions with soil abiotic properties (*Maestre et al., 2005*).

Arid soils have a particularly heterogeneous spatial distribution of abiotic properties (*Schlesinger et al., 1996*), particularly in nutrient content. Vegetation patches, deemed "fertility or resource islands," are also scarce and sparsely found in arid environments (*Cross & Schlesinger, 1999*; *Hirobe et al., 2001*; *Schade & Hobbie, 2005*). At the same time, there are large areas deprived of vegetation and severely limited in nutrients and water (*Evans et al., 2001*; *Belnap et al., 2005*), in which microbial communities, often referred to as "biological soil crusts" or "biocrusts," are the main drivers of energy input and biogeochemical processes (*Titus, Nowak & Smith, 2002*; *Belnap, 2003*; *Maestre et al., 2005*; *Housman et al., 2007*; *Castillo-Monroy et al., 2010*; *Bachar, Soares & Gillor, 2012*). Biocrusts contribute actively to natural small-scale soil heterogeneity, not only in terms of biological diversity but also in relation to soil function, including nutrient cycling and physicochemical properties associated with their spatial structure (*Maestre et al., 2005*).

Given the tight connection between microbial activity and nutrient cycling, it is reasonable to think that microbial distribution in soils might be somehow correlated with nutrients content across space (e.g., more nutrients, more microbial biomass and diversity). Despite the idea of resource island formation in arid soils, studies have shown that spatial distribution of microorganisms and nutrients is not correlated in these ecosystems (*Belnap et al., 2005*; *Housman et al., 2007*; *Geyer et al., 2013*). Some of these studies indicate that for arid, oligotrophic ecosystems, physicochemical parameters associated with water availability are better correlated with microbial distribution (*Geyer et al., 2013*). Thus, although there are some studies in arid ecosystems (*Barrett et al., 2006*; *Zeglin et al., 2009*; *Lee et al., 2012*; *Geyer et al., 2013*), it is of interest to gain better knowledge on the factors influencing microbial diversity, having direct consequences in soil fertility and ecosystem processes (e.g., soil mineralization and respiration rates) (*Maestre et al., 2005*; *Ben-David et al., 2011*).

In the present study, we aim to determine the spatial heterogeneity of microbial diversity and soil chemical parameters occurring in an arid soil of a hot desert ecosystem, in order to contribute with information and gain understanding into the aspects of the soil environment that are more strongly associated with differences in microbial community distribution in these kind of soils. We hypothesize that the spatial heterogeneity in chemical properties, previously reported for desert soils (*Schlesinger et al., 1996*), will be reflected in microbial diversity distribution at a yet unexplored local scale (order of meters). Thereby, in this study we: (a) characterize microbial community structure, (b) determine soil physicochemical and biochemical parameters and, (c) identify relationships among microbial community structure and chemical soil properties at a local spatial scale.

The study site, Cuatro Cienegas Basin (CCB), is located in a desert ecosystem in the middle of the Chihuahuan desert in Mexico. This is a gypsum-based system and is one of the most oligotrophic environments in the world. In contrast, the microbial diversity is very high in comparison to other arid soils (*López-Lozano et al., 2012*), providing the opportunity to investigate the spatial relationship between chemical distribution and microbial community structure, as it has been done in other oligotrophic arid ecosystems of Antarctica (*Zeglin et al., 2011*; *Geyer et al., 2013*).

## MATERIALS AND METHODS

### Study area
The study site is locally known as "Churince system." It is located in the western part of the CCB (26°50′N, 102°08′W; Fig. 1) at 740 m a.s.l. The system consists of a spring, an intermediate lagoon, and a dry desiccation lagoon connected by short shallow creeks. The annual precipitation in the area is less than 250 mm, occurring mainly from May to October. Temperatures fluctuate from 0 °C in January to 45 °C in July, with a mean annual temperature of 21.4 °C (CCB weather station). The vegetation is mainly halophile and gypsophile grasslands (*Challenger, 1998*). The area is also dominated by physical and biological soil crusts. The soil is predominantly basic, rich in calcium and sulfates, but very poor in nutrients, and belongs to Gypsisol type (*IUSS Working Group WRB, 2015*).

### Sampling design
The sampling plot was approximately at 50 m from the dry desiccation lagoon. The gypsophile grass *Distichlis spicata* covered 10% of the plot (Fig. 1) and it was the only species present in this area. Physical and biological crusts occupied the open areas between plants. In order to explore the spatial relationship between soil chemical distribution and microbial community structure at local scale, we used a plot of 8 m × 8 m that consisted of a nested system of four quadrats (*A–D* quadrats) of 16 m$^2$, which were divided in eight 1 m$^2$ "replicates," following a checkerboard pattern (Fig. 1). Vegetation cover for each sampling "replicate" or site (1 m$^2$) was registered qualitatively in order to have further ecological context for the results. In August 2007, we collected soil samples (500 g) from the first 10 cm at each site, to a total of 32 samples (eight samples for each 4 m$^2$ quadrat), under the SEMARNAT collection permits 06590/06 and 06855/07. Soil samples were homogenized in the field and divided in two subsamples, which were stored at −20 °C (for molecular
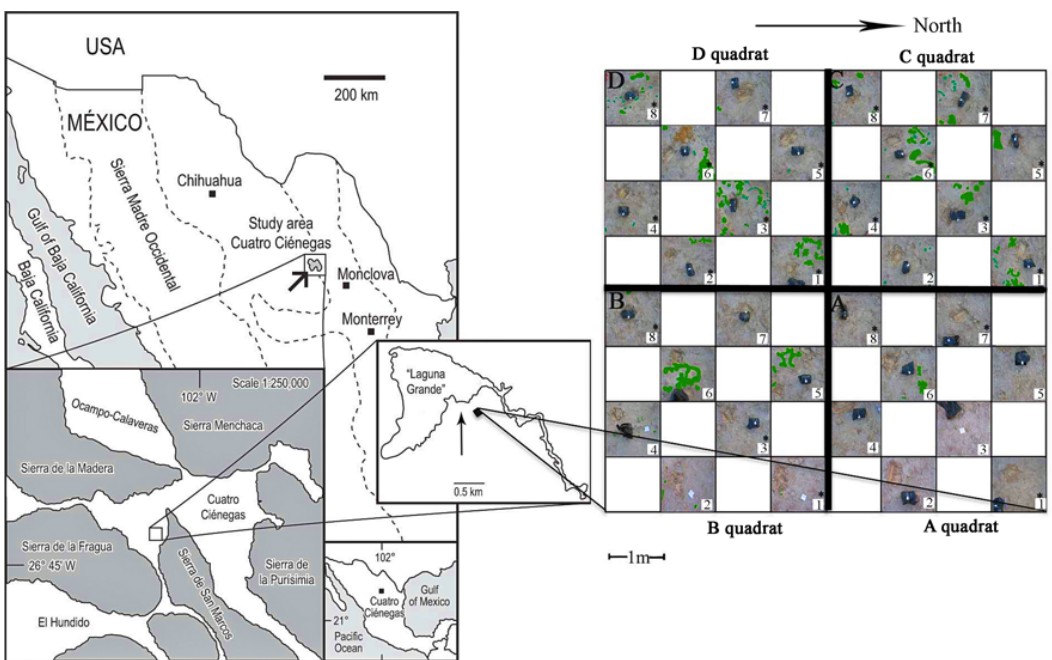

**Figure 1** **Sampling scheme.** An 8 × 8 m plot was selected in the Churince System within the Cuatro Cienegas Basin, México. A checkerboard sampling scheme was followed (*Noguez et al., 2005*) to a total of 32 samples, eight for each of the four quadrats (*A*, *B*, *C*, and *D*). Soil parameters were determined for the 32 samples. Numbers with asterisks indicate samples that were also analyzed for microbial diversity. Green colored areas indicate presence of vegetation.

analyses) and at 4 °C (for chemical analyses), respectively. Analyses were performed upon arrival to the laboratory.

## Physicochemical and biochemical analyses

Soil samples were air dried and sieved through a 2 mm mesh prior to physicochemical and biochemical determinations, which were performed twice for each sample. Total carbon (TC) was determined by dry combustion and coulometric detection (*Huffman, 1977*) using a Total Carbon Analyzer (TOC, UIC Mod. CM5012; UIC, Inc., Chicago, USA). For total nitrogen (TN) and phosphorus (TP), the samples were acid digested and determined colorimetrically using a Bran-Luebbe Auto-analyzer III (Germany), according to Bremner (*Bremmer & Mulvaney, 1982*) and Murphy & Riley (*Murphy & Riley, 1962*), respectively. Inorganic N forms ($NH_4^+$ and $NO_3^-$) were extracted with 2 M KCl after shaking for 30 min, followed by filtration through a Whatman #1 filter, and measured colorimetrically by the phenol–hypochlorite method. Inorganic P (Pi) was extracted with sodium bicarbonate, and determined colorimetrically by the molybdate-ascorbic acid method (*Murphy & Riley, 1962*). Dissolved organic C (DOC), N (DON) and P (DOP) were extracted with deionized water after shaking for 1 h and then filtered through a Whatman #42 filter. DOC was determined with a TOC module for liquids (UIC-Coulometrics), while DON and DOP were acid digested and measured colorimetrically.

Electrical conductivity and pH were determined in soil with deionized water (soil solution ratio 1:2). To quantify water-soluble cations ($Ca^{2+}$, $Mg^{2+}$, $K^+$, $Na^+$) and anions

$(HCO_3^-, Cl^-, SO_4^{2-})$, soil samples were shaken with deionized water for 19 h, centrifuged at 2,500 rpm and filtered through a Whatman #42 filter. $Ca^{2+}$ and $Mg^{2+}$ were analyzed by atomic absorption spectrophotometry with an air/acetylene flame (Varian SpectrAA 110), while $Na^+$ and $K^+$ by flamometry (flame photometer Corning PFP7) (*Bower, Reitemeier & Fireman, 1972*). Anions were determined by liquid chromatography (Waters Mod. 1525) with a mobile phase of borate sodium glucanate (*Bower, Reitemeier & Fireman, 1972*).

## Molecular analyses

Microbial community structure was characterized using terminal restriction fragment length polymorphisms (T-RFLPs) of bacterial 16S rRNA gene.

Genomic DNA was extracted from the soil samples using the Soil Master DNA Extraction Kit (Epicentre Biotechnology), with an additional previous step based on the fractionation centrifugation technique in order to reduce the high salt concentration (*Holben et al., 1988*). After extraction, genomic DNA was cleaned with Microcon columns (Fisher Scientific) with the purpose of removing any substance that could inhibit PCR amplification. These protocol modifications gave the best results from various methodologies tested; however, we were only able to amplify the 16S rRNA gene from 21 out of the 32 soil sampling sites (amplicons obtained in each quadrat: $A = 3$; $B = 3$; $C = 7$; $D = 8$). The low yield in the DNA amplification could be attributed to molecular applications inhibitors of unknown nature (*López-Lozano et al., 2012*).

Amplification of the bacterial 16S rRNA genes was carried out in a final volume of 50 μL containing: 0.2 μM of each fluorescently labeled domain-specific primers (VIC-27F and FAM-1492R) (*Lane, 1991*), 0.2 mM of each dNTP, 1 U of Taq Platinum DNA polymerase (Invitrogen), 2.5 μL DMSO, 2.5 μL BSA, 1 mM $MgCl_2$, 1 mM buffer, and 20 ng of DNA. Five independent PCR reactions were performed for each sample with the following program: 5 min at 94 °C; 30 cycles at 94 °C for 1 min, 52 °C for 2 min, 72 °C for 3 min; and 72 °C for 10 min. PCR products were pooled and purified from 2% agarose gel (Gel extraction kit, Qiagen Inc.). The amplicons were restricted with AluI enzyme (Promega) at 37 °C for 3 h and 65 °C for 20 min. Three independent readings of the size and abundance of fluorescently labeled terminal restriction fragments (TRFs) were performed for each sample using an ABI 3100-Avant Prism Genetic Analyzer (Applied Biosystems), as described previously (*Coolen et al., 2005*). For each profile of TRFs, we established a baseline and only those TRFs with peak heights ≥50 fluorescent units were used in subsequent analyses (*Blackwood et al., 2003*). Each unique TRF was considered to be an operational taxonomic unit (OTU). Estimations of diversity were derived from matrices constructed based on the presence and abundance of TRFs using relative peak area as an estimate of abundance calculated as:

$$Ap = (n_i/N) \times 100$$

in which $n_i$ represents the peak area of one distinct TRF and $N$ is the sum of all peak areas in a given T-RFLP pattern (*Lukow, Dunfield & Liesack, 2000*).

## Statistical analyses

Statistical and diversity analyses were performed in R (*R Development Core Team, 2011*), mainly with vegan (*Oksanen et al., 2012*), ggplots (*Warnes, 2012*) and BiodiversityR (*Kindt & Coe, 2005*) packages.

All soil properties data were expressed on a dry-weight basis. Non-normal data were log-transformed to normalize the distribution of the residuals. We analyzed the data using both multivariate (MANOVA to detect patterns: whether there were significant effects of the four quadrats on overall soil variables) and univariate (to detect significant differences in individual variables) methods. These analyses were followed by multiple pairwise tests, using Tukey's honestly significant difference (HSD), at the 5% level of significance, to identify possible differences in the soil variables between quadrats. The correlations between each pair of variables were calculated using Pearson's correlation coefficient. Soil properties were then standardized and ordered by principal components analysis (PCA), and the sampling points from the four quadrats were visualized with the two first principal components.

Alpha diversity indices (Shannon, Simpson, and Berger-Parker) and richness estimates were calculated for each quadrat using the T-RFLPs profiles. Microbial diversity indices were also analyzed using ANOVA type III for unbalanced data and evaluated using Renyi's entropy profiles for eight scales ($\alpha = 0$, 0.25, 0.5, 1, 2, 4, 8, infinite) (*Rényi, 1961*; *Chao et al., 2014*) with the BiodiversityR package. These profiles provide a comprehensive analysis of the diversity, giving a parametric measure of the uncertainty of predicting the OTUs richness, as well as the relative abundance of OTUs, at different scales between the four quadrats. To evaluate the sampling effort, rarefaction curves were constructed for each quadrat using EstimateS v.9.1.0 (*Colwell, 2005*). Microbial community structure between quadrats was examined through Venn diagrams (with the matrix of OTUs presence) and ordination analyses (with the matrix of OTUs abundance). To visualize communities' structure, Bray-Curtis dissimilarity distances were calculated with the relative abundance of T-RFLPs profiles. Similar communities were then clustered using the Ward's hierarchical clustering algorithm, which tries to minimize variances in agglomeration. A heatmap of the relative abundance of OTUs was constructed with dual hierarchical clustering.

Community structure was also investigated for correlations with chemical parameters following a multivariate analysis. For this, the relative abundance of T-RFLPs profiles were ordered by Detrended Correspondence Analysis (DCA) with Hellinger transformation (*Blackwood et al., 2003*), and correlations between the ordination axes and soil properties were calculated. This eigenvector-based ordination technique uses a chi-square distance measure and assumes that TRFs have a unimodal distribution along ecological gradients (*Legendre & Legendre, 1998*), which is a more appropriate assumption than linearity for ecological analysis of T-RFLPs data (*Culman et al., 2008*). Permutation tests under reduced model were used to identify significant explanatory soil variables ($p < 0.05$), which were plotted onto the ordination map as vectors. Only the soil variables corresponding to the same sampling sites as the T-RFLPs data were used for this analysis.

To further evaluate which soil abiotic properties could be used as predictors of the distribution of OTUs diversity, a multiple linear regression analysis was performed.

**Table 1 Physicochemical parameters.** Results from soil physicochemical analyses (mean ± standard deviation) of the four studied quadrats within Churince System in the Cuatro Cienegas Basin (Mexico).

| Variable | Quadrat | | | | Overall mean |
|---|---|---|---|---|---|
| | *A* | *B* | *C* | *D* | |
| Total C (mg g$^{-1}$) | 2.4 ± 0.8 | 2.4 ± 0.4 | 2.6 ± 0.4 | 2.8 ± 0.6 | 2.6 ± 0.6 |
| Total N (mg g$^{-1}$) | 0.57 ± 0.13 | 0.48 ± 0.18 | 0.59 ± 0.1 | 0.60 ± 0.18 | 0.56 ± 0.15 |
| Soil C:N | 4.6 ± 2.3 | 6.1 ± 4 | 4.6 ± 0.7 | 5.0 ± 1.3 | 5.1 ± 2.4 |
| Total P (mg g$^{-1}$) | 0.03 ± 0.01 | 0.03 ± 0.01 | 0.04 ± 0.01 | 0.04 ± 0.02 | 0.04 ± 0.01 |
| $NH_4^+$ (μg g$^{-1}$) | 4.0 ± 0.6 | 4.2 ± 0.8 | 4.0 ± 0.6 | 3.6 ± 1 | 4.0 ± 0.7 |
| $NO_3^-$ (μg g$^{-1}$) | 1.8 ± 1.9 | 1.5 ± 1.5 | 1.6 ± 1.7 | 2.3 ± 1.5 | 1.7 ± 1.6 |
| Dissolved organic C (μg g$^{-1}$) | 97.3 ± 27.8 | 75.8 ± 30 | 83.0 ± 33 | 124 ± 21.3 | 95.1 ± 33.2 |
| Dissolved organic N (μg g$^{-1}$) | 14.6 ± 3.4 | 18.1 ± 12.3 | 17.9 ± 8.7 | 19.6 ± 9.2 | 17.6 ± 8.7 |
| Dissolved organic C:N | 7 ± 2.9 | 6.4 ± 4.8 | 5.2 ± 2.4 | 7.9 ± 3.9 | 6.4 ± 3.5 |
| Dissolved organic P (μg g$^{-1}$) | 4.3 ± 3.6 | 5.6 ± 2.1 | 2.6 ± 3.5 | 3.3 ± 3.7 | 3.9 ± 3.3 |
| pH | 8.6 ± 0.1 | 8.7 ± 0.1 | 8.7 ± 0.1 | 8.8 ± 0.1 | 8.7 ± 0.1 |
| Electrical conductivity (dSm$^{-1}$) | 1.4 ± 0.1 | 1.4 ± 0.3 | 1.6 ± 0.2 | 1.4 ± 0.4 | 1.4 ± 0.3 |
| $Mg^{2+}$ (cmol kg$^{-1}$) | 27.1 ± 5.2 | 28 ± 6.3 | 35.2 ± 4.3 | 36.5 ± 4.5 | 31.7 ± 6.5 |
| $Ca^{2+}$ (cmol kg$^{-1}$)[*] | 0.56 ± 0.03[a] | 0.55 ± 0.02[a] | 0.64 ± 0.07[ab] | 0.65 ± 0.07[b] | 0.59 ± 0.07 |
| $Na^+$ (cmol kg$^{-1}$) | 140 ± 15.9 | 127 ± 16.1 | 166 ± 38.1 | 157 ± 28.4 | 147 ± 29.1 |
| $K^+$ (cmol kg$^{-1}$)[*] | 0.95 ± 0.14[a] | 0.82 ± 0.19[a] | 1.27 ± 0.14[b] | 1.33 ± 0.25[b] | 1.09 ± 0.28 |
| $HCO_3^-$ (cmol kg$^{-1}$)[*] | 2.8 ± 0.2[b] | 2.3 ± 0.6[b] | 1.2 ± 0.2[a] | 1.3 ± 0.3[a] | 1.9 ± 0.8 |
| $Cl^-$ (cmol kg$^{-1}$)[*] | 2.8 ± 0.2[b] | 2.5 ± 0.3[b] | 1.1 ± 0.4[a] | 1.3 ± 0.3[a] | 1.9 ± 0.8 |
| $SO_4^{2-}$ (cmol kg$^{-1}$)[*] | 15.1 ± 2.2[b] | 16.4 ± 2.5[b] | 7.6 ± 0.8[a] | 7.2 ± 0.9[a] | 11.6 ± 4.6 |

**Notes.**
Variable acronyms: C, carbon; N, nitrogen; P, phosphorous.
[*]Significant difference among quadrants ($p < 0.05$).
Different letters indicate that means are significantly different among quadrats.

Significant explanatory variables were chosen as the best predictors of OTU abundance by stepwise multiple regressions and by minimizing the Akaike Information Criterion (AIC). Collinearity between explaining variables was also tested using the Variance Inflation Factor (VIF).

## RESULTS

### Spatial heterogeneity in physicochemical and biochemical parameters

The total plant cover in the experimental plot was only 10%. However, quadrats *C* and *D* were more densely and homogeneously covered than *A* and *B* (Fig. 1). Overall there was a high presence of soil crusts and biocrusts, particularly in the *A* quadrat.

Soil chemical properties varied significantly between the four quadrats (Wilks' lambda = 0.000, $F = 7.896$, $p < 0.05$). Soil samples were alkaline (pH between 8.6–8.8) due to the high presence of salt in this arid ecosystem (Table 1). Cations ($Ca^{2+}$, $K^+$) and anions ($HCO_3^-$, $Cl^-$, $SO_4^{2-}$) were the most variable parameters in this small plot showing significant differences between quadrats, which means that ions significantly contribute to soil heterogeneity in this system. *C* and *D* quadrats had the greatest concentration of cations (0.64 and 0.65 cmol $Ca^{2+}$ kg$^{-1}$, 1.27 and 1.33 cmol $K^+$ kg$^{-1}$, respectively), except

for $Mg^{2+}$ and $Na^+$, while *A* and *B* quadrats had the highest concentration of anions (2.8 and 2.3 cmol $HCO_3^-$ $kg^{-1}$, 2.8 and 2.5 cmol $Cl^-$ $kg^{-1}$, 15.6 and 16.4 cmol $SO_4^{2-}$ $kg^{-1}$, respectively). The high concentration of $Na^+$ found in these soils (mean value of 147 cmol $kg^{-1}$) indicates salinity stress. Total forms and nutrients content were very low in this arid soil (TC: 2.4–2.8 mg $g^{-1}$; TN: 0.48–0.6 mg $g^{-1}$; TP: 0.03–0.04 mg $g^{-1}$; $NH_4^+$: 3.6–4.2 $\mu$g $g^{-1}$; $NO_3^-$: 1.5–2.3 $\mu$g $g^{-1}$), as expected, and they did not show significant differences among quadrats. The total C/N ratio (a quality index for soil organic matter) was also very low (from 4.6 to 6.1) and did not show significant differences between the four quadrats. On the other hand, C availability (DOC) was higher in *D* quadrat (124 $\mu$g $g^{-1}$), which means greater substrate availability for microbial metabolism in this quadrat. As expected, the pH was positively correlated with $Mg^{2+}$ and $Na^+$ (Table S1). The TC was only positively correlated with $Ca^{2+}$, while TP was positively correlated with pH and cations, as well as negatively with DOP. The TN was positively correlated with DON and negatively with C:N, $NH_4^+$, $NO_3^-$ and $HCO_3^-$. Finally, N inorganic forms were also positively correlated between them and the C:N ratio.

The complex chemical spatial heterogeneity among these four quadrats was explored using a PCA (Fig. 2). The first component (PC1) explained 54.3%, while the second component (PC2) explained 34.3% of the total variation in the soil parameters among quadrats (Table S2). The variables associated with the PC1 were cations and anions, as well as pH, TP, DOC and DOP. The PC2 was mainly related to soil nutrients (TN, C:N, $NH_4^+$, $NO_3^-$, DON, DOC:DON). A clear separation between quadrats was observed along the PC1 axis, mainly explained by the spatial heterogeneity distribution of ions.

## Spatial heterogeneity of microbial diversity

A total of 184 different OTUs were obtained in the four quadrats, of which 121 OTUs had less than 1% of the total maximum relative abundance. Unfortunately, the number of available samples was unbalanced for the microbial diversity study in this plot (amplicons in each quadrat: $A = 3$; $B = 3$; $C = 7$; $D = 8$) potentially due to the presence of inhibitors of unknown nature, which hampered the 16S rRNA amplification from all samples sites. Despite this constraint, rarefaction curves showed a good community sampling for quadrats *A*, *C* and *D*, with evident subsampling for quadrat *B*, which is one of the two quadrats for which only three out of eight samples could be analyzed in terms of microbial diversity (Fig. S1). Significant variation in alpha diversity indices among quadrats was detected. Although the *A* quadrat was also limited in the number of analyzed samples (3), it was the most diverse (*H*: 3.31) and with the highest evenness (1/*D*: 0.944; *BP*: 0.153), followed by *C*, *D*, and *B* (Table 2). It was also evident the high variability in microbial diversity among replicates (with the exception of *A*), which reflects the spatial heterogeneity at small local scale of this arid soil. A summary of diversity indices was obtained by calculating Renyi's community profiles. Ranking based on these profiles is preferred to ranking based on single indices because rank order may change when different indices are used (*Kindt & Coe, 2005*). These profiles showed the same pattern of diversity, both in terms of richness and evenness, where the highest diversity was found for quadrat *A* and the lowest for quadrat *B*, while *C* and *D* presented intermediate diversity (Fig. 3).

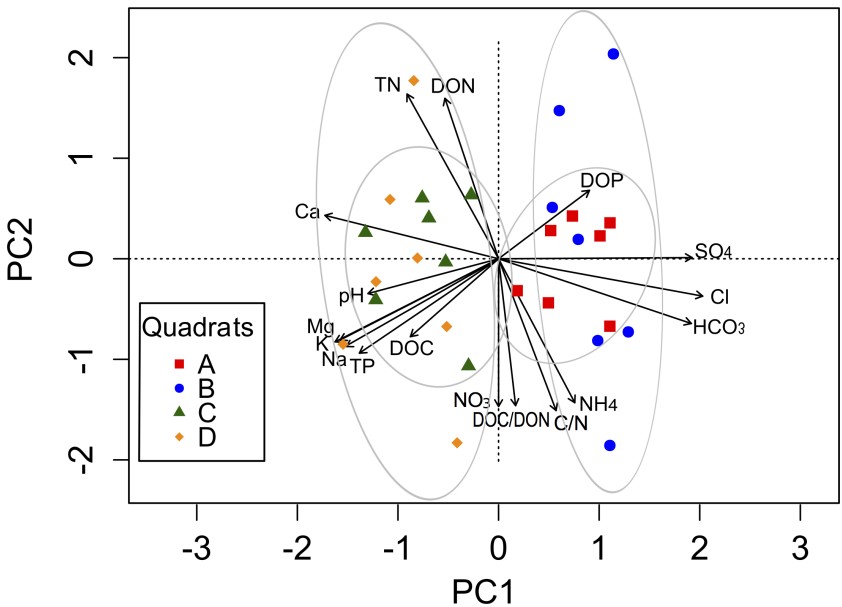

**Figure 2   Biplot generated from Principal Component Analysis (PCA) of the standardized soil variables for the four quadrats.** Symbols represent the different quadrats. Each vector points to the direction of increase for a given variable and its length indicates the strength of the correlation between the variable and the ordination scores. Ellipses show confidence intervals of 95% for each sample type. The first component of the PCA analysis accounted for 54.3% of the total variation, and the second component accounted for 34.3% of the variation.

**Table 2   Alpha diversity estimates.** OTUs diversity indices (mean ± standard deviation) from the T-RFLPs data of the four quadrats (*A*: 3 samples; *B*: 3 samples; *C*: 7 samples; *D*: 8 samples).

| Quadrat | Richness (S) | Shannon (H)[*] | Simpson (1/D)[*] | Berger–Parker[*] |
|---|---|---|---|---|
| *A* | 48 ± 9 | 3.31 ± 0.08[a] | 0.944 ± 0.004[a] | 0.153 ± 0.004[b] |
| *B* | 36 ± 15 | 1.98 ± 0.62[b] | 0.704 ± 0.145[b] | 0.497 ± 0.144[a] |
| *C* | 45 ± 13 | 2.56 ± 0.44[ab] | 0.8 ± 0.105[ab] | 0.393 ± 0.138[a] |
| *D* | 47 ± 19 | 2.3 ± 0.77[ab] | 0.738 ± 0.189[b] | 0.426 ± 0.203[a] |

**Notes.**
[*]Significant difference among quadrats ($p < 0.05$).
Different letters indicate that means are significantly different among quadrats.

Despite the high heterogeneity in microbial diversity in such a small plot, Venn diagram revealed a considerable overlap of OTUs among the four quadrats: 18% of OTUs were shared by all quadrats (Fig. 4), which represent 44.2% of the total abundance of the microbial community recovered from this plot with the T-RFLP technique. The *C* quadrat had the most "unique" OTUs (12%, representing 0.5% of total abundance), followed by quadrat *D* (11.4% representing 1.7% of total abundance), *B* (10.3%, representing 3.9% of total abundance) and *A* (5.4%, representing 0.3% of total abundance) in decreasing order. Interestingly, *C* and *D* quadrats shared 84 of the 184 recovered OTUs (46%), suggesting that both quadrats had more similar community composition than *A* and *B* quadrats. Moreover, the heatmap (Fig. S2) and the cluster dendrogram (Fig. S3) of OTUs abundance showed the same pattern of grouping as the PCA analysis for soil chemical properties,

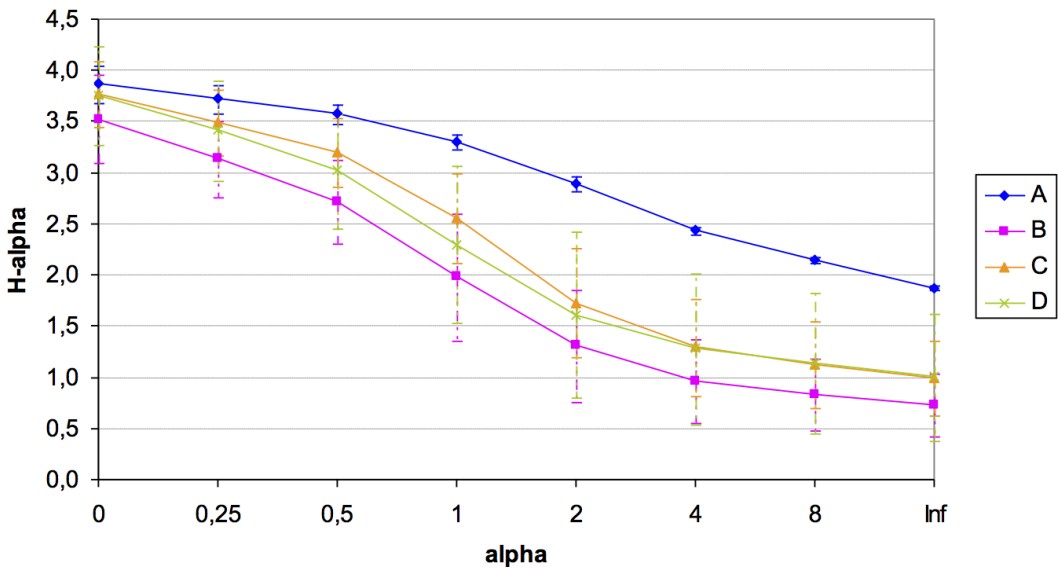

**Figure 3** Renyi's entropy profiles for the studied quadrats (*A*: 3 samples; *B*: 3 samples; *C*: 7 samples; *D*: 8 samples). Profiles were calculated with the OTUs abundance matrix. The alpha scale shows the different ways of measuring diversity in a community. Alpha = 0 is richness, alpha = 1 shows Shannon diversity, alpha = 2 is Simpson index (only abundant species are weighted), and alpha = Infinite only dominant species are considered (Berger–Parker index). The height of $H$-alpha values show diversity (for more information, see *Kindt & Coe, 2005*).

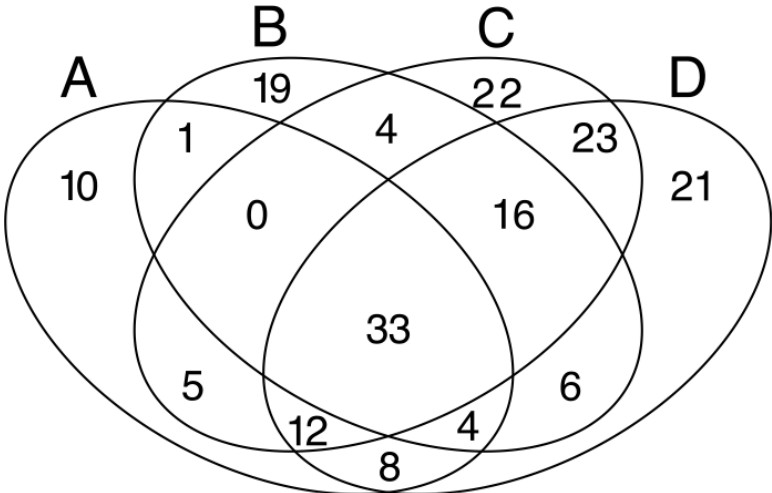

**Figure 4** Venn diagrams. Displaying the degree of overlap of OTUs composition among the four studied quadrats (A: 3 samples; B: 3 samples; C: 7 samples; D: 8 samples).

separating quadrats in two groups: *A*–*B* and *C*–*D*. As in the PCA analysis, there was a sample in the *C* quadrat that clearly deviated from the other samples.

## Multivariate analyses of microbial community structure

To explore the association between community structure and soil parameters, we performed a DCA analysis. The original 19 soil parameters were reduced to seven non-redundant

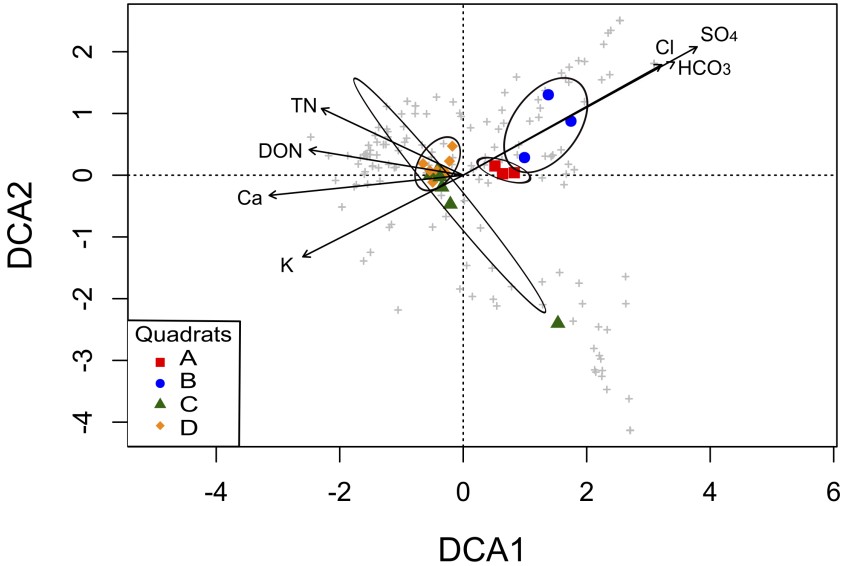

**Figure 5 Detrended Correspondence Analysis (DCA) of the T-RFLPs profiles with respect to the soil properties.** Sample sites for the four quadrats are represented by symbols, and OTUs are represented by grey crosses. Ellipses show confidence intervals of 95% for each sample type. Vectors stand for significant soil variables ($p < 0.1$). Each vector points to the direction of increase for a given variable and its length indicates the strength of the correlation with the axes.

explanatory variables (TN, DON, $Ca^{2+}$, $K^+$, $HCO_3^-$, $Cl^-$, and $SO_4^{2-}$; Table S3), which were the factors that contributed significantly to differences in community composition among the four quadrats in the ordination analysis. The DCA showed a clear separation of the quadrats in two groups, mainly explained by soil salinity: anions ($HCO_3^-$, $Cl^-$, and $SO_4^{2-}$) significantly correlated with OTUs from the *A* and *B* quadrats, while TN, DON, $Ca^{2+}$ and $K^+$ significantly correlated with OTUs from the *C* and *D* quadrats (Fig. 5). Thus, the grouping pattern of these microbial communities showed in the above analyses was also confirmed by the DCA analysis.

To further investigate the relation among OTUs diversity distribution and soil abiotic properties, a full linear model was then tested and reduced for a model where all the coefficients were statistically significant and the best AIC achieved. The best regression model indicated that DON, $Ca^{2+}$, $K^+$, and $SO_4^{2-}$ were statistically significant to explain the distribution of OTUs diversity in this soil (Table S4; $R^2 = 0.59$, $p < 0.01$).

## DISCUSSION

### Chemical heterogeneity at local spatial scale is mainly due to ions concentration variability

The values of TC, TN and TP in the experimental plot were lower than the values reported for other deserts (*Thompson et al., 2006*; *Strauss, Day & Garcia-Pichel, 2012*), as well as for soils in the CCB (*López-Lozano et al., 2012*; *Tapia-Torres et al., 2015*). The very low C:N ratio also suggests deficiency of soil organic C, therefore a low nutrient availability to soil microbes and vegetation, limiting the N cycle due to the lack of C availability. The Redfield

ratio in this soil was 71:17:1, which suggests that in these quadrats the C is the limiting nutrient in comparison with a general "average" soil C:N:P of 186:13:1 (*Cleveland & Liptzin, 2007*). On the other hand, this result also differs from the Redfield ratio of 104:5:1 reported for the same soil system (*López-Lozano et al., 2012*). These differences could be attributable to the great heterogeneity of this arid environment and the different time of soil sampling in both studies: February 2007 (dry cold season with low evapotranspiration) in *López-Lozano et al. (2012)* and August 2007 (rainy hot season with high evapotranspiration) in this study.

All soil samples in this study had high alkalinity produced by the elevated concentrations of ions, which is a general pattern in desert soils (*Titus, Nowak & Smith, 2002*). The high pH decreases P availability, which is very scarce in these soils and it is bonded to $Ca^{2+}$ and $Mg^{2+}$ (*Cross & Schlesinger, 2001*; *Perroni et al., 2014*). It is worth to mention that ions varied spatially in identity in this small plot: quadrats *A* and *B* were significantly high in anions, while quadrats *C* and *D* were significantly high in cations. The huge concentration of $Na^+$ in the four quadrats is an indicator of the extremely high salinity in these soils, which negatively affects the soil aggregates stability, as well as nutrients and water availability for plants, favoring the development of soil crusts, which are typical in arid and semiarid soils (*Belnap, 2003*; *Zhang et al., 2007*). In particular, salt crusts are abundant in the CCB area. They consist of layers at the soil surface mainly formed by soluble salt crystallizing soil particles at shallow saline groundwater level regions (*Zhang et al., 2013*).

The high concentration of ions can be attributed to the gypsum-rich nature of the CCB soils, where groundwater rises to the surface by soil capillarity action and water evaporation promotes salt accumulation. This situation results in rivers with a steep salinity gradient (*Cerritos et al., 2011*) and pools surrounded by saline soils rich in sulfates and extremely poor in nutrients (*López-Lozano et al., 2012*). Therefore, it is not surprising to find that the soil properties variation in this small plot was mainly explained by ions concentration, grouping the four quadrats in two broad clusters: *A–B* and *C–D*. These clusters had a qualitative pattern associated with the vegetation cover, being quadrats *C* and *D* more densely and homogeneously covered by vegetation than *A* and *B*. Although the present research analyzes soil communities, a previous study of microbial communities of the water system associated with the studied plot showed a correlation of microbial composition and water conductivity gradients (*Cerritos et al., 2011*). Thus, the spatial variation in these physicochemical properties among the four quadrats may be a consequence of differences in moisture content due to the proximity to a subterranean water flow, indirectly evidenced by the marked patchy distribution of the vegetation cover and the "open" areas occupied by soil crusts (*López-Lozano et al., 2012*).

## Heterogeneity in microbial diversity at local spatial scale is explained by physicochemical factors, not by vegetation cover nor nutrient content

Despite recent important advances in our knowledge of the structure, composition and physiology of biotic components in arid soils (*Belnap et al., 2005*; *Caruso et al., 2011*; *Maestre et al., 2015*; *Makhalanyane et al., 2015*), little is known about the spatial variability

of microbial diversity at local scales and its interactions with chemical heterogeneity in these ecosystems (*Housman et al., 2007*; *Castillo-Monroy et al., 2011*; *Andrew et al., 2012*). T-RFLPs fingerprinting was used in this study to assess the relationship between microbial structure and the small-spatial heterogeneity of soil chemical properties. We are aware that this technique cannot recognize taxonomic groups and accounts mainly for relatively abundant microbial groups. Nevertheless, given the aims of this study of characterizing microbial communities structure and its relationship with abiotic or physichochemical parameters, a fingerprint approach such as T-RFLPs is adequate to provide replicable, valid and sufficient data (*Angel et al., 2013*), as it has done for many other studies looking at patterns of correlation between microbial diversity/composition and environmental factors (*Fierer & Jackson, 2006*).

The heterogeneity in OTUs diversity among these quadrats is evident, being the *A* quadrat the most different with respect to the other quadrats. Despite the fact that the *A* quadrat had scarce plant cover and similar nutrients and ions concentrations to the *B* quadrat, it showed the greatest microbial diversity, which could be related to the high presence of biocrusts that may incorporate more resources to the soil, potentially increasing organic C and biomass and, in turn, diversity (*Geyer et al., 2013*). On the other hand, the *B* quadrat had the lowest microbial diversity, which could be related to the lowest values of DOC found in this quadrat. Labile organic matter fractions, such as DOC, are the primary energy source for soil microorganisms and are characterized by rapid turnover (*Bolan et al., 2011*). It has been reported that even in disturbed sites, DOC is the main source of C influencing the composition of the microbial community (*Churchland, Grayston & Bengtson, 2013*). Then, changes of soil microbial community could be regulated by C availability through labile soil organic matter pools, as have recently been shown to happen in McMurdo Dry Valleys arid soils in Antarctica (*Geyer et al., 2013*).

Regarding similarity in microbial composition among the four quadrats, cluster dendrogram and multivariate analyses showed two clear groups, which were *A–B* and *C–D*, corresponding to the same clustering of quadrats based on soil chemical parameters. A common explanation for the soil microbial composition patterns is related to the presence of plants controlling levels of microbial diversity and driving community assembly (*Singh et al., 2007*; *Berg & Smalla, 2009*; *Ben-David et al., 2011*). However, in our study the observed spatial pattern of microbial diversity distribution at such local scale does not seem to be associated with vegetation cover. For example, the *A* quadrat is the most diverse in microbial community and the less vegetated, suggesting that microbial diversity in this arid soil could be more related to the presence of "open" areas occupied by biocrusts. On the other hand, abiotic factors, such as ionic content, are statistically explanatory variables in the spatial ordering (DCA) of the microbial communities analyzed. In addition, the multiple regression model selected shows the importance of DON, $Ca^{2+}$, $K^+$, and $SO_4^{2-}$ in explaining the distribution of OTUs abundance in this arid soil. Abiotic drivers of microbial diversity in arid soils has been also reported for the Sonoran desert (*Andrew et al., 2012*), where location, pH, cation exchange capacity and soil organic C were highly correlated with microbial composition. Therefore, we showed that microbial community diversity

and distribution responds to and/or influences local soil physicochemical characteristics at a small spatial scale in this arid ecosystem.

## CONCLUSIONS

In desert areas such as CCB, soil moisture is one of main limiting factors affecting vegetation growth and distribution, as well as soil microbiology. The gypsum-based water system controls the soil physicochemical factors and ultimately the microbial community distribution in this arid ecosystem. Thus, the high heterogeneity in the soil properties and microbial community among these small four quadrats seems to be a consequence of differences in the soil saline content. In addition, the high concentration of $Na^+$ favors the emergence of both salt and biological crusts and the irregular plant cover distribution in this system. Local spatial variability of physicochemical properties and microbial diversity observed in this arid ecosystem is likely to exist in most soils ecosystems, and needs to be considered when making ecological inferences and when developing strategies to sample the soil environment. A better understanding of the role of spatial heterogeneity in biotic and abiotic factors will help to determine the relevance of small-scale studies for large-scale patterns and processes.

## ACKNOWLEDGEMENTS

We want to thank Dr Laura Espinosa-Asuar and Rodrigo González Chauvet for technical support during the development of this study.

### Funding

This work was supported by the Universidad Nacional Autónoma de México (UNAM-PAPIIT grant IA200814 to AEE) and SEMARNAT-CONACyT (grant 23459 to VS). SP received financial support from a CONACyT research visiting fellowship (186372). The funders had no role in study design, data collection and analysis, decision to publish, or preparation of the manuscript.

### Grant Disclosures

The following grant information was disclosed by the authors:
Universidad Nacional Autónoma de México: IA200814.
SEMARNAT-CONACyT: 23459.
CONACyT research visiting fellowship: 186372.

### Competing Interests

Valeria Souza and Luis E. Eguiarte are Academic Editors for PeerJ.

### Author Contributions

- Silvia Pajares conceived and designed the experiments, analyzed the data, wrote the paper, prepared figures and/or tables, reviewed drafts of the paper.

- Ana E. Escalante conceived and designed the experiments, analyzed the data, wrote the paper, reviewed drafts of the paper, contributed reagents/materials/analysis tools.
- Ana M. Noguez conceived and designed the experiments, performed the experiments, analyzed the data, wrote the paper, reviewed drafts of the paper.
- Felipe García-Oliva, Luis Enrique Eguiarte and Valeria Souza conceived and designed the experiments, contributed reagents/materials/analysis tools.
- Celeste Martínez-Piedragil performed the experiments.
- Silke S. Cram performed the experiments, contributed reagents/materials/analysis tools.

## Field Study Permissions

The following information was supplied relating to field study approvals (i.e., approving body and any reference numbers):

The field permit for biological sample collection was granted by the Environmental Council in Mexico (SEMARNAT) to Valeria Souza: Permit numbers 06590/06 and 06855/07.

## Data Availability

The raw data has been supplied as Data S1.

## Supplemental Information

Supplemental information for this article can be found online at http://dx.doi.org/10.7717/peerj.2459#supplemental-information.

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
