# Peer review of "Spatial heterogeneity of physicochemical properties explains differences in microbial composition in arid soils from Cuatro Cienegas, Mexico"

_PeerJ, doi:10.7717/peerj.2459_

## Round 0.1 · original submission · Major Revisions

· Academic Editor

Major Revisions

The manuscript was reviewed by 3 authors, while 2 consider minor revisions, Reviewer 1 is concerned about the limited number of samples and small area, 32 soil samples within a 64 m^2 plot. The reviewer is also concerned the way the data were analysed. The authors are required to address all of these issues in the revision.

Also note the annotated manuscript from Reviewer 2.

Reviewer 1 ·

Basic reporting

The manuscript by Pajares et al. uses a very standard methodological approach that aims at correlating observed patterns of microbial community structure (or composition) with the soil metadata. By reading through this manuscript I identified two major drawbacks that hamper the scientific appreciation and validation of the findings and conclusions, as follows: (i) the low resolution of the approach. It’s well-known to date that the resolution obtained by fingerprinting methods is far below the state of the art that is current possible with high-throughput sequencing methods. I’m not stating that such classical methods must be abandoned, but I’m favor of their application in more simplified systems (i.e. those containing 10 to 100 OTUs), rather than in a soil sample. On top of that, in the methods section (L164), the authors state that TRF profiles were converted in presence-absence matrices. Why to do that? In doing so, all the data on relative abundance of particular T-RFs are lost. I wonder how the authors continued on the analyses and made inferences about community diversity and evenness. At least as it is stated, it is only possible to infer on communities richness at most. This sum with the confusion on the choice of the distance matrix (Bray-Curtis, L182) – this is not appropriated for presence-absence matrices. Moreover, the authors claim in the abstract that they analyzed 32 soil samples (which is very limited), but in fact they succeed to analyze only 21 (L148), which is even a reduced number. (ii) the writing style. The manuscript is confusing and does not give the impression that the arguments are building up a coherent/linear story. I found many points that can be improved, but those mentioned below are far from be the complete list.

Experimental design

As mentioned above the experimental design is very limited. Taken together these weaknesses hinder the realistic appreciation of the data. The limitation goes from the number of samples to the methods applied. It is also not clear the choice for the experimental design (i.e. the 64m2 plot). The authors must give a better justification for their design.

Validity of the findings

As it stands the data provided are very weak to support the arguments. As previously mentioned, how possible is to make inferences on community diversity and evenness based on presence-absence data? As it stands the manuscript is methodologically wrong and cannot support the claims. Moreover, with such a reduced dataset (only 21 samples) and limited number of analyzed TRFs (<100) the results are likely to be strongly biased.

Comments for the author

Abstract
From line 28 to 30. There is an overuse of the word ‘heterogeneity’. Please revise it accordingly.
Title and L35. Not clear what exactly is ‘ions content’.
L38. The correct is 16S rRNA gene, and in this case bacterial 16S rRNA gene.
L40 and all over the manuscript. Why the use of the term ‘biogeochemical properties’? when only the soil chemical properties were determined. There is no ‘bio’ in the analyses. And as far as I understood the physical structure of the soils were not either determined.
L42. Why to evoke biocrusts here if not introduced before? Not clear.

Introduction
L45. ‘functional implications’, please describe which ones.
L68-L72. This whole section on biocrusts is disconnected from the text, please revise it accordingly.
L73-L82. This paragraph is too vague. What is the main assumptions here? Revise for clarity.
L92-L97. This paragraph can be moved to the Materials and Methods section.

Materials and Methods.
L149-L151. How well is supported this argument. There was no validation or experimental test for such claim. Remove it or revise it.
L164. ‘presence-absence matrices’. See comments above.

Results.
This whole section needs more ‘meet on the bones’. In other words, the statements are too vague and not supported by numbers. This section must be revised and numbers (the real data) must be incorporated along the sentences. As it stands this section is a combination of personal findings and feelings on the data. The point here is to let the data speaks by itself.
L226. The fact that the analyzed quadrats contain disproportional number of samples can impose statistical bias. As more samples may likely lead to more diverse measurements. I strongly advise for data rarefication/subsampling to the minimum number across quadrats. This must, at the very least, be provided as supplementary material.
L226-L227. Very speculative sentence.

Discussion.
This section is often too speculative and moves beyond what the findings can support. Please revise it carefully.
L307-L312. These arguments are not enough to justify the limitation of the approach, see my comments above.
L313-L317. This is wrong. Clone libraries are also very limited in resolution. To assume that a soil community is composed by only a few hundred OTUs is very risk. I doubt that this is the case for these samples. Please remove or revise the sentence accordingly.
L324-L329. These arguments are out of topic. There is no data on community composition in the manuscript.
Conclusions.
Remove the first sentence. All studies are unique in their own topic. No need to state that.

·

Basic reporting

The manuscript fits most of the "Basic Reporting" requirements. I have highlighted a few areas in my copy of the .pdf where some editing of language is necessary. These were all very minor however. The structure of the article is clear and well thought-out. I believe Figure 3 is somewhat unclear and the Renyi's profile procedure should be better described in the main paper. Also, Figure 5 (dendrogram) is unnecessary. Standard deviation ellipses could be used in either or both ordinations (Fig 2, 6) around each quadrant to highlight the grouping of A+B versus C+D, thus replacing Fig. 5.

Experimental design

I disagree that the affect of geochemistry on microbial diversity is "unexplored" as the authors state in the Introduction. I have provided a number of citations in the .pdf for important research that has been done in the polar deserts of Antarctica (McMurdo Dry Valleys), including some research that has compared polar deserts to the Chihuahuan Desert.

Validity of the findings

Overall the findings are well supported by the data and evenly discussed. I would bring up two points: 1) why was soil moisture not measured for these soils? This value may show strong variability among the sampled quadrants, particularly between vegetated and unvegetated soils and likely drives both soil chemistry and microbial diversity. 2) The use of univariate ANOVA to measure significant differences among the 19 response variables in Table 2 is inappropriate. Multivariate ANOVA (MANOVA) would help to account for the multiple comparisons and adjust the p-value to avoid Type I errors.

Comments for the author

With the above changes this will be a nice manuscript that adds to the story of microbial biogeography and characterization of arid systems.

Reviewer 3 ·

Basic reporting

No Comments

Experimental design

I suggest to add rigorously all methods original reference and equipment details (model, factory).
Line 108, i suggest checking form and updating the reference to "IUSS Working Group WRB, 2007"

Validity of the findings

No Comments

Comments for the author

The manuscript presents interesting results, but the mix of statistical analysis is sometimes tangled and how they relate is not clear. The experimental design and investigation are rigorous, discussions are deep and relevant conclusions.

---

## Round 0.2 · Minor Revisions

· Academic Editor

Minor Revisions

Please make remaining minor revisions according to the comments of Reviewer 2.

·

Basic reporting

I see all of the Basic Reporting policies having been met.

Experimental design

The experimental design, as presented, is mostly sound. A few comments that still linger for me: 1) multicollinearity among the driving soil factors is not well described or controlled for in the analysis and 2) a multiple linear regression approach would help to resolve multicollinear concerns and would leave the reader with a much clearer idea of the main conclusions from this study.

It appears that many of the soil ions are strongly correlated with one another, which makes a determination of which variables are truly driving microbial diversity rather difficult. The description of using DCA mentions that only 7 non-redundant explanatory variables are used; does this imply they are uncorrelated? How were these variables chosen?

A robust way to handle the multicollinearity would be multiple linear regression (MLR). MLR will condense and rank the explanation of microbial diversity by different soil factors, while removing any that are strongly collinear. The results of this test would provide a more clear picture for the reader which soil factors are the most important drivers. As the manuscript stands, it is not clear which factors are most important. Instead, the Discussion focuses broadly on how microbial diversity differs between the four quadrants, which (in turn) have different chemistries and plant cover. This could be made more explicit via regression.

Validity of the findings

The conclusions, as they stand, are robust. However, see my comments above for suggestions to more rigorously examine the data.

Comments for the author

This manuscript includes valuable data that will be appreciated by the broader soil community. I encourage you to refine your Methods and Discussion, however, to include some regression results that could more closely define which soil factors are driving microbial diversity. Additionally, I have edited this draft of the manuscript with a few more grammatical corrections.

Reviewer 3 ·

Basic reporting

No Comments

Experimental design

No Comments

Validity of the findings

No Comments

Comments for the author

Following the changes performed by the autors, now the manuscript is in condition for publication in this journal. The study is clear, relevant and represents an advance in this scientific field.

---

## Round 0.3 · accepted · Accept

· Academic Editor

Accept

The authors have revised and addressed the reviewer's comment.